# A Secreted Form of the Hepatitis E Virus ORF2 Protein: Design Strategy, Antigenicity and Immunogenicity

**DOI:** 10.3390/v14102122

**Published:** 2022-09-26

**Authors:** Zihao Chen, Shaoqi Guo, Guanghui Li, Dong Ying, Guiping Wen, Mujin Fang, Yingbin Wang, Zimin Tang, Zizheng Zheng, Ningshao Xia

**Affiliations:** 1State Key Laboratory of Molecular Vaccinology and Molecular Diagnostics, National Institute of Diagnostics and Vaccine Development in Infectious Diseases, School of Public Health, Xiamen University, Xiamen 361005, China; 2School of Life Sciences, Xiamen University, Xiamen 361005, China; 3United Diagnostic and Research Center for Clinical Genetics, School of Medicine & School of Public Health, Xiamen University, Xiamen 361005, China; 4NMPA Key Laboratory for Research and Evaluation of Infectious Disease Diagnostic Technology, School of Public Health, Xiamen University, Xiamen 361005, China; 5Xiang An Biomedicine Laboratory, Xiamen University, Xiamen 361005, China; 6Research Unit of Frontier Technology of Structural Vaccinology, Chinese Academy of Medical Sciences, Xiamen 361005, China

**Keywords:** hepatitis E virus, ORF2, antigenicity, immunogenicity, B-cell decoy

## Abstract

Hepatitis E virus (HEV) is an important public health burden worldwide, causing approximately 20 million infections and 70,000 deaths annually. The viral capsid protein is encoded by open reading frame 2 (ORF2) of the HEV genome. Most ORF2 protein present in body fluids is the glycosylated secreted form of the protein (ORF2^S^). A recent study suggested that ORF2^S^ is not necessary for the HEV life cycle. A previously reported efficient HEV cell culture system can be used to understand the origin and life cycle of ORF2^S^ but is not sufficient for functional research. A more rapid and productive method for yielding ORF2^S^ could help to study its antigenicity and immunogenicity. In this study, the ORF2^S^ (tPA) expression construct was designed as a candidate tool. A set of representative anti-HEV monoclonal antibodies was further used to map the functional antigenic sites in the candidates. ORF2^S^ (tPA) was used to study antigenicity and immunogenicity. Indirect ELISA revealed that ORF2^S^ (tPA) was not antigenically identical to HEV 239 antigen (p239). The ORF2^S^-specific antibodies were successfully induced in one-dose-vaccinated BALB/c mice. The ORF2^S^-specific antibody response was detected in plasma from HEV-infected patients. Recombinant ORF2^S^ (tPA) can act as a decoy to against B cells. Altogether, our study presents a design strategy for ORF2^S^ expression and indicates that ORF2^S^ (tPA) can be used for functional and structural studies of the HEV life cycle.

## 1. Introduction

Hepatitis E virus (HEV) is the leading cause of enterically transmitted acute hepatitis worldwide, causing approximately 20 million infections and 70,000 deaths annually [1,2]. HEV infection is usually self-limiting, and chronic HEV infection has been described in immunosuppressed patients. Approximately 30% mortality of HEV infection has also been reported in pregnant women in the third trimester [3]. The diverse *Hepeviridae* family contains two HEV subfamilies, *Orthohepevirinae* (infecting terrestrial mammals and birds) and *Parahepevirinae* (infecting fish) [4]. The subfamily *Orthohepevirinae* includes four genera: *Paslahepevirus*, *Rocahepevirus*, *Chirohepevirus*, and *Avihepevirus*. At least six genotypes (gt) are pathogenic in humans and fall under the species *Paslahepevirus balayani* and *Rocahepevirus ratti* (https://ictv.global/report/chapter/hepeviridae/hepeviridae (accessed on 1 June 2022)) [5,6,7,8]. HEV gt1 and gt2 (*Paslahepevirus balayani)* occur mostly in resource-poor countries and regions and infect humans exclusively [9,10]. However, gt3, gt4, and gt7 of the species *Paslahepevirus balayani* and gt1 of the species *Rocahepevirus ratti* are zoonotic and can cause persistent infections in immunocompromised patients [6,11,12].

HEV is a single-stranded, positive-sense, and quasi-enveloped RNA virus, and the ~7.2 kb full-length genome expresses three open reading frames (ORFs) [1]. ORF1 is a nonstructural polyprotein indispensable for HEV replication. ORF2, comprising 660 amino acids (a.a.), encodes at least two forms of the ORF2 capsid protein: the secreted form of ORF2^S^ and the capsid-associated ORF2^C^ [13,14]. The glycosylated ORF2^S^ dimer was first reported by Montpellier et al. in 2018 [14]. Soon after, Yin et al. reported that translation from the first AUG of the ORF2 genome would produce ORF2^S^ protein with a signal peptide that could rapidly direct the protein into the secretory pathway without intracellular accumulation, whereas translation from the internal AUG would disrupt the signal sequence, producing a cytoplasmic ORF2^C^ protein that is responsible for virion assembly, which is not translocated into the endoplasmic reticulum or the Golgi apparatus and lacks the glycosylation modification [13]. In addition, ORF2^S^ is the major antigen in hepatitis E patient serum. ORF2^S^ does not prevent HEV entry but impairs antibody-mediated neutralization, so it has been speculated that ORF2^S^ is likely to act as an immunological decoy [13,14]. ORF3 is a small multifunctional protein related to viral morphogenesis and egress [15].

HEV is present as a naked virion in feces and bile and an enveloped virion in the serum of HEV-infected patients and HEV-infected cell culture systems [14,16]. A recent study described an efficient HEV cell culture system that led to an improved understanding of the HEV life cycle [14]. The accurate sequence of least two forms of the ORF2 proteins were identified through studies of this cell culture system. Relative to the ORF2 sequence, the first a.a. of ORF2^C^ and ORF2^S^ corresponds to Leu^14^ and Ser^34^, respectively, and the last a.a. corresponds to Ser^660^. Although the system has enabled increased expression of viral proteins and infectious particles, allowing direct biochemical analysis of the ORF2 proteins, the study of the immunological function, biological significance, and structure of ORF2^S^ is limited by the protein yield. A rapid and productive system for obtaining ORF2^S^ remains to be developed.

In our study, we describe a method for the expression of the ORF2^S^ protein that permits rapid recombinant expression and purification of ORF2^S^ protein at approximately 5 mg/L, permitting biological and structural research. We demonstrate a substantial antigenic overlap between ORF2^S^ and other HEV capsid proteins and that ORF2^S^ is immunogenic in mice. An excellent ORF2^S^-specific antibody response was induced in plasma from patients infected with HEV, and ORF2^S^ could serve as a decoy to allow the study of the B-cell response against HEV.

## 2. Materials and Methods

### 2.1. Construction of Four ORF2^S^ Expression Clones

HEV genome open reading frame 2 (ORF2) full-length sequence (a.a. Met^1^-Ser^660^) of genotype (gt) 4 (strain subtype 4a) was obtained from a male with chronic hepatitis E with aplastic anemia (GenBank no. OP185389). Then, the original secretion signal peptide of the HEV ORF2 (a.a. Met^1^-Gly^23^) was replaced by the tPA signal peptide. The tPA signal peptide and ORF2^S^ genes (a.a. Ser^34^-Ser^660^) were cloned into the pTT5 expression vector and named ORF2^S^ (tPA). Then, 10× His-tags were added to the C-termini of constructed genes. The HEV p239 (p239) genes of gt4 from a swine HEV gene (GenBank no. GQ166778) were cloned into the pTO-T7 plasmid [17].

### 2.2. Expression, Identification and Purification

The ORF2^S^ (tPA) construct was transiently transfected into Expi-293F cells, in accordance with the instructions of the manufacturer (Life Technologies, Carlsbad, CA, USA). The target proteins (ORF2^S^) were mainly secreted into the supernatants. Cell pellets were also collected for analysis. Cell supernatants and pellets were tested for HEV antigen (Ag) by a commercial ELISA kit (Wantai Biological Pharmacy Co., Beijing, China). The HEV Ag commercial ELISA kit was used for detecting the 10× His-tag of ORF2^S^ with 5000-fold diluted secondary horseradish peroxidase (HRP)-conjugated anti-His mAb (Proteintech, Chicago, IL, USA). Target proteins were separated by SDS/PAGE and subsequently blotted onto a nitrocellulose membrane (Whatman, Dassel, Germany). Mouse monoclonal anti-HEV antibody 1B7 was applied as the primary antibody (diluted 5000-fold), which recognized the linear epitope HEV ORF2 a.a. 443–457 [18]. HRP-conjugated goat anti-mouse mAb was used as the secondary antibody (Sino Biological Inc., Beijing, China, diluted 10,000-fold). Color was developed using SuperSignal West Femto Maximum Sensitivity Substrate (Thermo Scientific, Waltham, MA, USA). Cell supernatants were dialyzed into 20 mM phosphate buffered saline (PBS, pH = 7.4) at 4 °C overnight. The cell pellets were resuspended in 20 mM PBS and disrupted by sonication to produce lysates. The proteins in the cell supernatants and pellets were purified using a Ni Sepharose Excel column (GE Healthcare, Chicago, IL, USA) with 250 mM imidazole.

### 2.3. SDS/PAGE

ORF2^S^ proteins mixed with nonreducing and reducing 6× loading buffer were subjected to SurePAGE, Bis Tris, 8–16% gradient SDS-polyacrylamide gel electrophoresis (GenScript, Piscataway, NJ, USA). ORF2^S^ proteins were stained with Coomassie blue, in accordance with standard laboratory protocols.

### 2.4. HPLC

Purified HEV ORF2^S^ was subjected to high performance liquid chromatography (HPLC) (Agilent Technologies 1200 series; Santa Clara, CA, USA) through a TSK Gel G3000 pwxl 7.8 × 300-mm column (TOSOH, Tokyo, Japan) equilibrated in 20 mM PBS. The column flow rate was maintained at 0.5 mL/min, and the run time was 30 min. Purified HEV ORF2^S^ in PBS was detected at OD_280nm_.

### 2.5. Epitope Identification by a Mouse mAb Panel against ORF2^S^

Ninety-six-well microplates were coated with 100 ng/well of E2, p239, ORF2^S^ of HEV gt4 and p495 of gt1 overnight at 4 °C. The wells were blocked with 0.5% (*w*/*v*) casein in PBS at 37 °C for 2 h. As described in a previous report [18,19], 15 mouse monoclonal antibodies (mAbs) were sequentially distributed into 9 groups, referred to as C1 (8E1 and 3B11), C2 (8G12), C3 (5H6 and 12E11), C4 (6H8 and 10E11), C5 (3G3, 6F8, 12A7 and 8C11), C6 (9F7), L1 (15B2), L2 (12A10), and L3 (1B7). Representative mouse mAbs were diluted to concentrations of 20 μg/mL. Five-fold serially diluted mAbs were added to the microplates and incubated at 37 °C for 30 min. After 5 washes, HRP-conjugated goat anti-mouse IgG secondary antibody, diluted 5000-fold (Thermo Scientific, Waltham, MA, USA), was added to the microplates. After incubation at 37 °C for 30 min, the microplates were washed 5 times, and 100 μL of TMB was added to the wells. The reaction was stopped by adding 50 μL of 2 M H_2_SO_4_ after incubation at 37 °C for 15 min. The absorbance was measured at OD_450nm_ with a reference wavelength of OD_630nm_.

### 2.6. Animal Experiment

Nine special pathogen-free (SPF) female BALB/c mice of 6 weeks of age were randomly assigned into 3 groups of three BALB/c mice each. SPF BALB/c mice were vaccinated subcutaneously with a single dose of HEV gt4 of p239 and ORF2^S^ (tPA) vaccine, which had been mixed with Freund’s Complete Adjuvant (Sigma-Aldrich, St. Louis, MO, USA). The vaccine was manufactured in 20 μg formulation mixed with Freund’s complete adjuvant. Mice immunized with PBS suspended in Freund’s complete adjuvant were used as controls. Serum samples were collected weekly and tested for anti-HEV IgG at −1, 1, and 2 weeks of vaccination by indirect ELISA according to standard laboratory protocols. Briefly, 96-well microplates were coated with 100 ng/well of p239 and ORF2^S^ of HEV gt4. Serum diluted 10-fold was used as the first sample, and the anti-HEV IgG level was determined from 5-fold serially diluted serum samples. The median effective dose (ED_50_) was calculated based on ELISA data from postimmunization BALB/c mice.

### 2.7. Serology Testing of Natural HEV Infection

Fifty-five HEV-infected patients gave informed consent to participation in the study. This study was designed in accordance with the Declaration of Helsinki and subsequently approved by the medical ethics committee of the School of Public Health, Xiamen University. Peripheral blood was collected from patients naturally infected with HEV, and plasma was separated by centrifugation. WHO standard serum was used as a reference to quantify the titer of anti-HEV IgG in naturally infected patients [20]. The Anti-HEV IgG titer in the plasma was tested using 96-well microplates coated with 100 ng/well of p239 and ORF2^S^ of HEV gt4, as described in a previous method [19]. Briefly, plasma diluted 10-fold was used as the first sample. Five-fold serially diluted plasma was added to microwell plates and incubated at 37 °C for 30 min. After five washes, HRP-conjugated anti-human IgG antibodies were added. After incubation at 37 °C for 30 min, the plates were washed five times and 100 μL of TMB was added to the wells. The reaction was stopped after incubation at 37 °C for 15 min by adding 50 μL of 2 M H_2_SO_4_.

### 2.8. Protein Labeling and Flow Cytometry

ORF2^S^ (tPA) was separately labeled with DyLight 488 NHS Ester (Thermo Scientific, Waltham, MA, USA) and EZ-Lin Sulfo-NHS-LC-Biotin (Thermo Scientific, Waltham, MA, USA), in accordance with the instructions of the manufacturer. Peripheral blood mononuclear cells (PBMCs) were separated by Ficoll-Hypaque gradient (GE Healthcare, Chicago, USA) centrifugation. PBMCs were obtained from HEV 239 Hecolin (Xiamen Innovax, Xiamen, China) vaccinated donors 1 month after completing three vaccine doses according to a standard schedule (0, 1, 6 months) and were incubated with fluorescent antibodies and ORF2^S^ (tPA) for the identification of ORF2^S^-specific B cells. PBMCs were stained with a cocktail consisting of LIVE/DEAD Fixable Aqua Dead Cell Stain Kit (Invitrogen, Carlsbad, CA, USA, 1 per 100), CD3-PE-Cy7 (BD Biosciences, San Diego, CA, USA, 1 per 100), CD19-BV786 (BD Biosciences, San Diego, 1.5 per 100), CD27-PE (BD Biosciences, San Diego, 0.5 per 100), IgM-PerCP-Cy5.5 (BD Biosciences, San Diego, 2 per 100), IgG-BV421 (BD Biosciences, San Diego, 1.5 per 100), and the recombinant ORF2^S^ (tPA)-DyLight 488 (1 μg/μL) and ORF2^S^ (tPA)-Biotin (1 μg/μL) described above. Two consecutive staining steps were conducted. The first was Aqua/CD3/CD19/CD27/IgM/IgG/ORF2^S^ (tPA)-DyLight 488/ORF2^S^ (tPA)-Biotin in 100 μL PBS applied at 4 °C for 30 min. The second was streptavidin-allophycocyanin (Invitrogen, Carlsbad, USA, SA-APC) for an additional 30 min at 4 °C to target the biotin tag of ORF2^S^ (tPA). The stained cells were washed, resuspended in PBS, and strained through 70-μM Fisherbrand Sterile Cell strainers (Fisherbrand, former Savant, USA). ORF2^S^ (tPA)-specific IgG+ memory B cells were gated as Live/CD3^−^/CD19^+^/CD27^+^/IgM^-^/IgG^+/^ORF2^S+^.

## 3. Results

### 3.1. Construction, Expression, Identification, and Purification of ORF2^S^ Clones

The ORF2^S^ protein expression construct is shown in Figure 1A. The ORF2^S^ gene sequence was obtained from a chronic gt4a HEV-infected male patient with aplastic anemia (GenBank no. OP185389). The construct was named according to the expression vector and secretion signal peptide. The ORF2^S^ (tPA) construct was transiently transfected into Expi-293F cells. To estimate the yields and features of ORF2^S^, the supernatants and pellets of expressing cells were collected for analysis. HEV Ag and His tag were detected in the supernatants and pellets. Given the high sensitivity and specificity of commercial HEV Ag ELISA (Wantai, Beijing, China), the kit was used for detecting ORF2 protein levels [21,22,23,24,25]. The supernatants from ORF2^S^ (tPA) showed high levels of HEV Ag (Figure 1B), with a signal to cutoff (S/CO) ratio of up to 123,424, higher than those in the pellets (S/CO = 78,913). The anti-His tag reaction activity (median effective dose, ED_50_) in ORF2^S^ (tPA) supernatants (ED_50_ = 858.6) was also stronger than that in pellets (ED_50_ = 6.2) (Figure 1B). We performed Western blotting (WB) to characterize the proteins from the supernatants and ORF2^S^ (tPA) pellets. Boiled ORF2^S^ protein was detected as a monomer of approximately 90 kD in supernatants of ORF2^S^ (tPA) (Figure 1C), which is consistent with previous reports [13,14]. Whereas a smaller monomer of approximately 75 kD was detected in pellets of ORF2^S^ (tPA), which suggests that the ORF2^S^ (tPA) in pellets lacks posttranslational modifications. Further glycosylation analysis confirmed that only the ORF2^S^ (tPA) protein in the supernatant was glycosylated, and the small protein in the pellet did not undergo posttranslational modifications (Figure 1D). The ORF2^S^ dimer in previously reported efficient HEV cell culture systems were glycosylated protein [13,14]. The posttranslational modifications of ORF2^S^ protein in supernatants suggest that ORF2^S^ (tPA) supernatants are better candidates than pellets for purification. The supernatants of ORF2^S^ (tPA) were harvested and purified by Ni-excel affinity chromatography. SDS/PAGE and WB showed that most of the ORF2^S^ protein, corresponding to dimers of approximately >180 kD and monomers of approximately 90 kD, was eluted by 250 mM imidazole (Figure 1E). To examine the purity of the ORF2^S^ proteins, gel filtration high-performance liquid chromatography (HPLC) analysis was performed. A single peak (11.6 min) demonstrated the presence of ORF2^S^ (tPA) (Figure 1F), suggesting successful purification from expressing cell supernatants.

### 3.2. Epitope Features of ORF2^S^ and Immunogenicity Assessed in BALB/c Mice

To assess the antigenic differences between glycosylated dimeric ORF2^S^ protein and ORF2^C^ protein lacking posttranslational modifications, we performed a detailed epitope mapping, using a panel of mAbs that recognize nine different antigenic epitopes (C1–C6 and L1–L3) to compare the antigenicity of different forms of HEV capsid proteins. The panel of mAbs used was generated in a previous study [19]. The HEV E2s domain (a.a. 459–606) was the major target in the antibody response to HEV and was divided into six distinct clusters (C1-C6), as previously reported [19]. C1 epitopes were located in the protrusion (P) domain (a.a. 452–606) at the bottom region of the dimer. The C2 epitope was located around the dimerization interface. The C3 and C4 epitopes were located on top of the P domain. The C5 epitope was within the groove zone of the dimerization interface, and the C6 epitope was located around the groove zone [19]. C2, C3, C5, and C6 are the major neutralizing epitopes in the E2s domain. C3 and C4 are associated with receptor binding [26]. Linear epitopes L1, L2, and L3 were located at a.a. 403–417, a.a. 423–437, and a.a. 443–457 of the middle (M) domain (a.a. 318–451), respectively [18,26]. The encoded a.a. positions of four proteins, their aggregate forms, the three definite domains of the HEV capsid protein, and the epitope regions recognized by the panel of mAbs are depicted in Figure 2A. This mapping revealed substantial antigenic overlap between ORF2^S^ and other HEV ORF2^C^ capsid proteins, including p495 (a.a. 112–606), p239 (a.a. 368–606), and E2 (a.a. 394–606) (Figure 2B). Linear epitopes L1-L3 were not different in the four HEV capsid proteins (Figure 2B). The most obvious difference among the capsid proteins was the loss of the antigenic epitope C3 on ORF2^S^. The glycosylation of key residues in ORF2^S^ may mask antigenic epitope C3, which is located at the top region of the P domain (Figure 2B). To verify this result, glycosidase digestion (PNGase F and Neuraminidase A) was performed. The ORF2^S^ protein was sensitive to digestion with PNGase F and neuraminidase A, demonstrating that it is an N-glycoprotein and a sialylated protein (Appendix A). In addition, we performed alignment of the ORF2 a.a. 394–606 sequences of these four proteins. In the alignment, we identified one potential glycosylation site (Asn^562^, as previously reported [14]) that may affect the binding of C3 antibodies to the ORF2^S^ protein, that is, one potential glycosylation residue in ORF2^S^ may mask antigenic epitope C3 (Figure 2C). Since the gene encoding the p495 protein is HEV gtA1, sequence differences at sites (Gly^490^ and Asn^490^) between HEV gt1 and gt4, which are recognized by C4 antibodies, resulted in decreased binding (Figure 2C).

Given that the ORF2^S^ antigen is abundant in the serum of HEV-infected patients, the immunogenicity of purified ORF2^S^ in BALB/c mice was investigated. ORF2^S^ (tPA) and ORF2^C^ (p239) were tested as candidates for investigation of the immunogenicity and immunogen-induced antibody response. Groups of three animals each were immunized with one dose of a 20 µg formulation of ORF2^S^ (tPA), p239, or PBS, and serum samples were collected weekly. The immunization program is shown in Figure 3A. According to the overlapping antigenicity, ORF2^S^ and p239 proteins were tested separately for anti-HEV IgG by indirect ELISA. The serum conversion rate of anti-HEV IgG could be stabilized at 100%, as detected by the increased antibody response to the two proteins after immunization (Figure 3B). Intriguingly, testing on microplates coated with p239 protein revealed similar anti-HEV IgG titers for the p239 (ED_50_ = 25.51) and ORF2^S^ (tPA) (ED_50_ = 23.15) immunized groups, whereas distinct titers (ED_50_ = 0.64 versus 4.24) were shown for the ORF2^S^ (tPA) protein (Figure 3C). ORF2^S^ (tPA) proteins were immunogenic, and ORF2^S^ (tPA)-specific antibodies were successfully induced in the one-dose-vaccinated BALB/c mice.

### 3.3. Analysis of ORF2^S^-Induced Antibody Response

To validate ORF2^S^ as an immunological decoy to modulate antibody responses, we collected blood samples from 55 patients infected with HEV, and plasma was separated by centrifugation. Anti-HEV IgG titers were quantified using serial plasma dilutions on ELISA plates coated with either p239 or ORF2^S^ (tPA). Plasma showed significantly higher anti-HEV IgG titers binding to ORF2^S^ (tPA) (mean = 23.43 WU per mL) than p239 (mean = 3.04 WU per mL) (Figure 4A). The increased antibody titers may be related to the presence of more antigenic epitopes on ORF2^S^ (tPA). Further analysis suggested a significant correlation between anti-HEV IgG titers binding to ORF2^S^ and p239 (*p* < 0.0001, R^2^ = 0.252) (Figure 4B). Although anti-HEV IgG titers binding to ORF2^S^ (tPA) showed an overall increase compared with those of p239, the titers decreased in some samples (Figure 4C). Of the 55 plasma samples, the majority (83.64%) showed increased anti-HEV IgG titers binding to ORF2^S^ (tPA), while the remainder (16.36%) showed decreased titers, suggesting excellent antigenicity of ORF2^S^ (tPA) binding to plasma from HEV-infected patients. ORF2^S^ is present in body fluids and is the major HEV ORF2 protein in HEV-infected patient serum. Induced antibodies mainly target the ORF2^S^ protein. However, substantial antigenic overlap was observed between ORF2^S^ and ORF2^C^ (that is, p239), so a small subset of antibodies also targets the p239 protein. These data suggest that ORF2^S^ may act as an immunological decoy to induce antibody response in patient serum. To further investigate this possibility, we examined ORF2^S^-induced B-cell responses at the cellular level. Flow cytometry was used to assess B cells that recognized the ORF2^S^ (tPA) probe. The gating strategy for assessing ORF2^S^ (tPA)-specific memory B cells is shown in Appendix A. An obvious ORF2^S^ (tPA)-specific CD27^+^ IgG^+^ B-cell response was revealed by flow cytometry. The proportions of ORF2^S^ (tPA)-responding B cells varied between vaccinated and naturally infected donors (Figure 4D). The levels in HEV-infected individuals (ranging from 0.260% to 1.030%) were significantly higher than those in HEV-vaccinated individuals (0.082%) (Figure 4E). These results demonstrated that ORF2^S^ (tPA) aroused an antigen-specific adaptive humoral response and that the immunogen may induce antibodies that recognize ORF2^S^-distinct epitopes located in regions outside p239 (a.a. 368–606). In addition, based on serum antibody levels and B-cell responses, it appears that ORF2^S^ plays an immunomodulatory role to induce an antibody response. Since ORF2^S^ does not prevent HEV entry but impairs antibody-mediated neutralization, the ORF2^S^ protein could serve as an additional mechanism by which virions evade host immunity.

## 4. Discussion

Here, we generated an ORF2^S^ (tPA) expression construct as an efficient tool to express and purify the HEV ORF2^S^ protein. Protein identification experiments with ORF2^S^ (tPA) showed that the ORF2 protein was mostly secreted into supernatants, as previously described, and the supernatant expression level of the ORF2 protein was S/CO 123,424 (approximately 5 mg/L after purification) [13,14,16,27]. We demonstrated that ORF2^S^ (tPA) from supernatants has a relatively complete a.a. sequence, whereas C-terminally truncated ORF2 protein was detected in cell pellets. The secreted ORF2^S^ (tPA) proteins was eluted by 250 mM imidazole and corresponded to a glycosylated monomeric protein of approximately 90 kD, consistent with the results of a previously reported cell culture system [13,14]. In a previous study, Yin at al. demonstrated that dimerization of the ORF2^S^ protein does not require disulfide bonds formation, as is the case for the ORF2^C^ protein [13]. The mechanism of ORF2^S^ protein dimerization is currently unknown. The structure of the E2s domain of the ORF2^C^ protein shows that Ala^597^, Val^598^, Ala^599^, Leu^601^, and Ala^602^ are directly involved in the dimerization of ORF2^C^ protein [28]. Therefore, it will be necessary in the future to identify whether these a.a. are also involved in ORF2^S^ dimerization. Importantly, the dimeric form of the ORF2^C^ protein is essential for its HEV neutralizing function [28]. Hence, we speculate that the dimeric form of ORF2^S^ is essential for its potential function. Further research is needed to test this hypothesis.

Epitope mapping, using a panel of mAbs that recognize nine different antigenic epitopes [18,19], revealed substantial antigenic overlap between ORF2^S^ and three other HEV ORF2^C^ capsid proteins: p495, p239, and E2. Although ORF2^S^ shares a consistent nucleotide identity with ORF2^C^, antigenic epitope C3 was lost in ORF2^S^ (tPA). The reason for the loss of the C3 epitope may be that it is masked at the one potential glycosylation site (Asn^562^); alternatively, there may be a difference in conformation between ORF2^S^ and ORF2^C^ [29]. Future research will clarify the sources of epitopes differences and overlap between ORF2^S^ and ORF2^C^, and additional studies are also needed to examine the biological function of ORF2^S^.

Our HPLC analyses showed that ORF2^S^ (tPA) is stable and homogenous in PBS, with a peak at 11.6 min. In contrast to previous low-yield HEV-infected cell cultures [30,31], the production of ORF2^S^ (tPA) at approximately 5 mg/L was sufficient for structural analysis, allowing the elucidation of several unrecognized biological aspects of ORF2^S^. In addition, the sites of glycosylation could be clearly revealed by the structural analysis. Thus, we provided a design strategy for future experiments to express the ORF2^S^ protein and carry out structural studies.

Similar but not identical epitope profiles were recognized by a panel of mAbs, including previously detected E2 domain epitopes C1–C6 and linear epitope regions L1–L3 [13]. We found that antigenic epitope C1 was weakened, and C3 was lost in ORF2^S^ (tPA). We did not detect the loss of antigenic epitope C4 with the same antibody used in the Yin at al. study [13]. The difference in recognition by antigenic epitope C4 is likely attributable to the low protein capture in the HEV-infected cell culture system. Yin et al. suggested that the loss of antigenic epitopes C3 and C4 was likely responsible for the lack of repression of HEV virions in target cells by ORF2^S^, although Montpellier et al. [14] revealed that ORF2^S^ carries a lower capacity for HEV virions formation. Further studies are needed to elucidate this issue.

During its life cycle in HEV-infected patients, the secreted form of the ORF2^S^ protein circulates in body fluids (serum and urine) and is the predominant form of ORF2 protein in the serum. We found that serum from HEV-infected patients showed significantly increased titers of anti-HEV IgG binding to ORF2^S^ (tPA) compared with p239 protein. Furthermore, ORF2^S^ (tPA) aroused an antigen-specific antibody response in the BALB/c model, and our data indicate that an obvious ORF2^S^ (tPA)-specific B-cell response was present in HEV-infected patients. ORF2^S^ impairs antibody-mediated neutralization, as demonstrated by Yin et al. through an in vitro neutralization experiment, suggesting that ORF2^S^ suppressed the effect of neutralizing mAbs [13]. Our data support the theory that ORF2^S^ serves as an immunological decoy and, thereby, plays a role in blocking antibody neutralization. However, the immunoregulatory action of the ORF2^S^ protein is unknown. It would be interesting to determine whether ORF2^S^-protein-induced monoclonal antibodies recognize HEV virions.

Recent studies have defined the origin and identification of ORF2^S^, which HEV-infected cell cultures secretes into supernatants. Given the low-yield of these cultures, we present an efficient design strategy for expressing the ORF2^S^ protein. Complete epitope mapping of the protein was also performed. The ORF2^S^-induced antibody response was detected in the BALB/c mouse model and in plasma and B cells from HEV-infected patients. In summary, this study will contribute to further research on HEV ORF2^S^ protein structure and biological function.

## Figures and Tables

**Figure 1 viruses-14-02122-f001:**
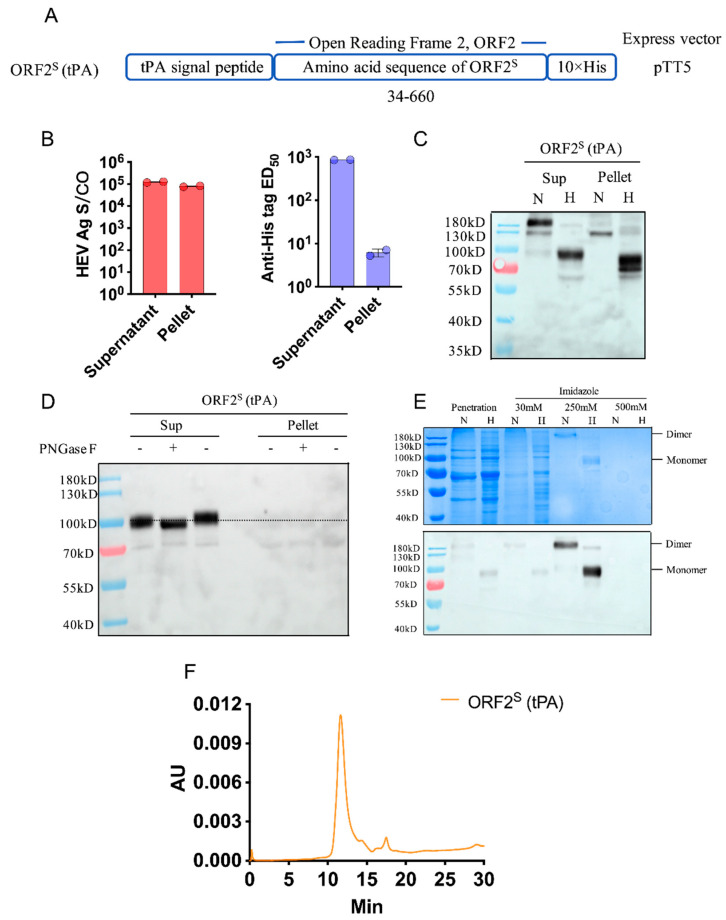
Design strategy, expression, identification, and purification in constructed ORF2^S^ clone. (**A**) Expi-293F cells were transfected with the ORF2^S^ (tPA) constrict. Mass spectrometry indicated that a.a. 34–660 sequence of ORF2 corresponds to the ORF2^S^ protein. Then, 10× His-tags were added to the C-termini of the construct genes. Cell supernatants and pellets were collected for further analysis. (**B**) Given the high sensitivity and specificity of commercial HEV Ag ELISA (Wantai, Beijing), the kit was used to detect the ORF2 protein levels in supernatants and pellets. HEV Ag levels are shown as the signal to cutoff (S/CO) ratio. The original #4 secondary antibody (Wantai, Beijing, China) in the kit was replaced with an anti-His tag secondary antibody (Proteintech) to detect the 10× His tag. Anti-His tag levels are shown as the median effective dose (ED_50_). (**C**) Supernatants and pellets were collected and analyzed by Western blotting (WB) with mouse monoclonal antibody 1B7, which recognizes the linear epitope HEV ORF2 a.a. 443–457. (**D**) Glycosylation analysis of ORF2^S^ (tPA) proteins in supernatants and pellets. ORF2^S^ (tPA) proteins in supernatants and pellets were denatured and digested with indicated glycosidases (+) or without (−). The dashed line shows the mobility shift on nitrocellulose filter membrane of ORF2^S^ (tPA) proteins to assess the extent of de-glycosylation. (**E**) Components of the imidazole-eluted concentration gradient were analyzed by SDS/PAGE and WB. The majority of ORF2^S^ was eluted by 250 mM imidazole solution. (**F**) Gel filtration high-performance liquid chromatography (HPLC). Purified HEV ORF2^S^ (tPA) was equilibrated in 20 mM PBS and detected in PBS at OD_280nm_. The column flow rate was maintained at 0.5 mL/min, and the run time was 30 min. N, no heat under nonreducing conditions; H, heat under reducing conditions.

**Figure 2 viruses-14-02122-f002:**
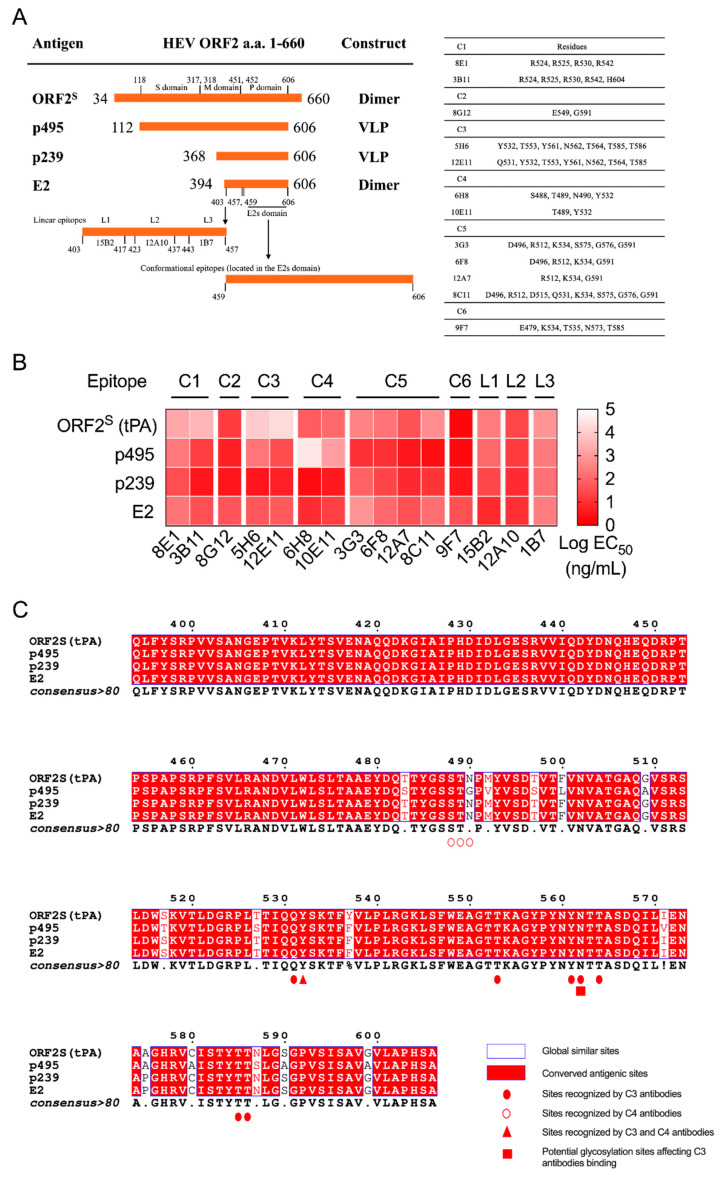
Comparison of immunoreactivity and epitope mapping in different HEV capsid proteins (ORF2^S^, p495, p239, and E2). A panel of anti-ORF2 monoclonal antibodies (mAbs) that recognize nine different antigenic epitopes (C1-C6 and L1-L3) was used for detection. (**A**) Information on the four proteins and epitope regions recognized by a panel of mAbs (adapted with permission from Ref. [A Comprehensive Study of Neutralizing Antigenic Sites on the Hepatitis E Virus (HEV) Capsid by Constructing, Clustering, and Characterizing a Tool Box]. 2015, Zhao et al.). S domain, shell domain; M domain, middle domain; P domain, protruding domain; VLP, virus-like particles. (**B**) The panel of mAbs used was generated in a previous study [19]. Indirect ELISA was performed with a goat anti-mouse antibody (Thermo Scientific) as the visualized secondary antibody. The data shown are the concentrations that elicit 50% of maximal effect (EC_50_). C1–C6, Cluster 1–Cluster 6; L1–L3, linear epitope 1–linear epitope 3. (**C**) Alignment of amino acid sequences (394–606) of four HEV capsid proteins (ORF2^S^, p495, p239, and E2). The blue frame indicates globally similar sites. The red box indicates antigenic sites conserved among the four proteins. Red solid circles, red hollow circles, and red solid triangles represent sites recognized by C3 antibodies, C4 antibodies, and both C3 and C4 antibodies, respectively. Red solid rectangles represent potential glycosylation sites affecting C1 antibody binding.

**Figure 3 viruses-14-02122-f003:**
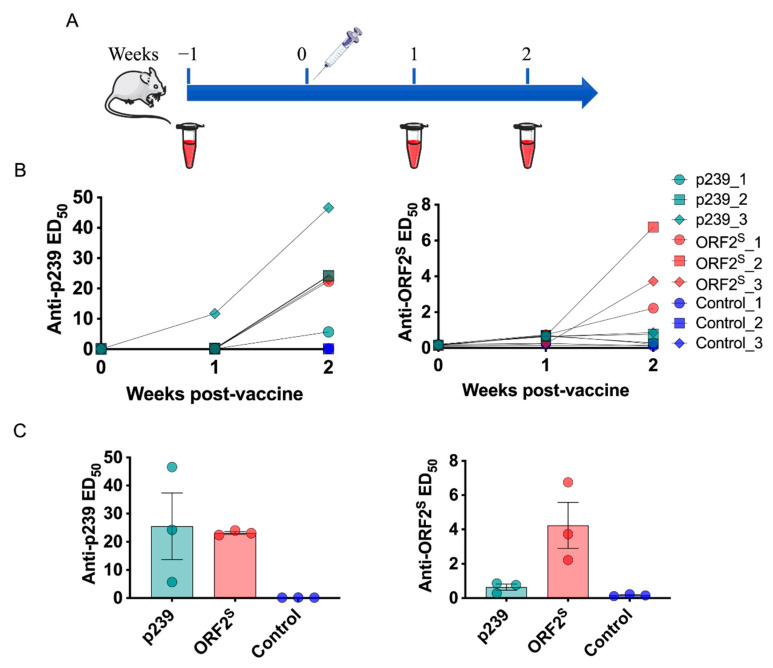
ORF2^S^ as an immunogen for assessing immunogenicity. (**A**) Immunization schemes of the animal experiments. BALB/c mice were administered one dose of 20 µg ORF2^S^ (tPA), p239, or PBS, and serum samples were collected weekly. (**B**) Serum anti-HEV IgG titers over time. The ORF2^S^ (tPA) and p239 proteins of HEV gt4 were tested separately. The data are shown as the median effective dose (ED_50_). (**C**) Comparison of anti-HEV IgG serum titers in mice at the second week. Black bars indicate the standard error of the mean (SEM). The column data show the mean with SEM.

**Figure 4 viruses-14-02122-f004:**
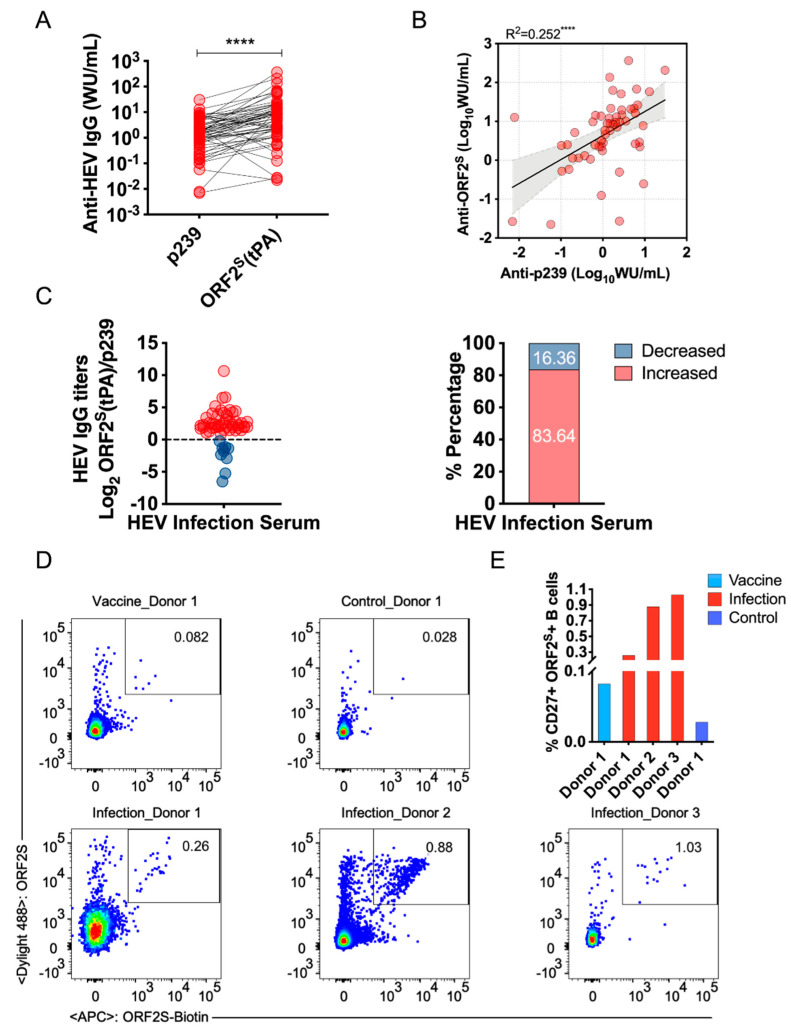
Analyses of the antibody response to ORF2^S^ in plasma and B cells. (**A**) Serial dilutions of plasma samples separately bound to ORF2^S^ (tPA) and p239. The WHO standard serum was used as a reference to quantify the titer of anti-HEV IgG in naturally infected patients. The black lines are drawn between the same plasma samples. ****, *p* < 0.0001. (**B**) Correlation of anti-HEV IgG titers measured for ORF2^S^ (tPA) (Y axis) and p239 (X axis). The black line indicates the fitted linear regression. The 95% confidence intervals (CI) are shown in silver. ****, *p* < 0.0001. (**C**) Comparison of anti-HEV IgG titers between ORF2^S^ and p239. Solid red dots indicate increased anti-HEV IgG titers, and solid blue dots indicate decreased anti-HEV IgG titers binding to ORF2^S^ (tPA), compared with p239. Data for anti-HEV IgG titers binding to ORF2^S^ (tPA)/p239, converted to log2. (**D**) Calibration with nonspecific memory B cells using control cells from healthy donors (top right). The proportions of ORF2^S^-specific memory B cells are recognized by two fluorescently labeled ORF2^S^ (tPA) probes (labeled with DyLight 488 and allophycocyanin (APC), respectively). Recognized memory B cells from HEV 239-vaccinated donors are shown at the top left. ORF2^S^-specific memory B cells from HEV-infected donors are highlighted in boxes (bottom). (**E**) The proportions of ORF2^S^-specific memory B cells are displayed as columns. Light blue, red, and dark blue columns indicate vaccinated, naturally infected, and control donors, respectively.

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
