# Peer review of "A Secreted Form of the Hepatitis E Virus ORF2 Protein: Design Strategy, Antigenicity and Immunogenicity"

_viruses, 2022, doi:10.3390/v14102122_

Round 1

Reviewer 1 Report

In this manuscript, the authors presented an efficient way to express and purify the secreted form of HEV ORF2 protein, of which the immunogenicity was then characterized. This is an interesting study, but the manuscript is not suitable for publication in its current format.

General comments:

The language is a major problem of this manuscript. It is really not clear what the authors want to express and why things have been done. Thorough polishing is absolutely necessary.

For examples (among many others):

Line 143: „can case persistent infections” should be “can cause persistent infections”

Line 160: What is an expression clone?

Line 168: “expression clone” should be “expression construct”

Line 171: A verb is missing in this sentence.

Line 224: “Thus, ORF2s may serve as a candidate to be used for researcher”, I am not sure what the authors want to convey here. 

Line 292: This sentence is just terrible.

Line 304: “Researcher would be clarified sources of different antigenic epitope”-What does this mean?

Specific comments:

Introduction:

  • Introduction section should go before the materials & methods section.
  • Much more information on the different ORF2 forms is required. Both forms are secreted-just along different pathways. The ORF2s form is glycosylated, this should be mentioned and addressed somewhere. It is speculated in the literature already that this form acts as an immunological decoy. The entire study should be put within this context and clear aims and biological questions formulated. 

Figure 1:

  • Only single data point? No repeats?
  • Line 183: I do not understand how the authors concluded that the supernatant is a better source of ORF2s. Again, the aim of producing and isolating this ORF2 form is not well explained in the introduction section. Why is the supernatant better than the intracellular recovered one? Because of proper post translational modification? Here, clear distinguishment between infectious and secreted/glycosylated ORF2 should be made and analyzed (e.g. PNGase F treatment). What does N & H stand for in this figure? No explanations were found anywhere in the manuscript, neither the materials & methods section nor the figure legend. 

Figure 2:

  • Add an introductory sentence that the mABs used in this study were generated in a previous study by the same group.
  • A scheme of ORF2 with the different mAB epitopes, with respect to the ORF2 M, S, and P domains as well as the glycosylation sites, would be useful to clarify how purified ORF2s react to different antibodies.

Figure 3:

The purpose of immunizing mice with ORF2s is not well explained. 

Figure 4: 

It seems that the authors wanted to show that the ORF2s has translational potential, but it should be decently explained and put into the right scientific context. Also, the question remains to clarify the biological significance of these data. 

Discussion:

In general, the discussion is really superficial. Entire discussion about monomers vs dimers is unclear. Does the protein act as an immunological decoy or not? In vitro neutralization data would be good to assess this. 

Author Response

Dear reviewer,

Thanks to the editor for arranging the review. Thanks to the reviewers very much for your review of the article "A secreted form of the hepatitis E virus ORF2 protein: design strategy, antigenicity and immunogenicity", we have replied the questions raised by reviewers. All changes in the manuscript are highlighted in yellow. Because of your suggestions, the revised manuscript gets better and could get more valuable information for reader. It is hoped that this version of the manuscript can meet the requirements of reviewer.

Yours sincerely

Reviewer 2 Report

In this manuscript, the authors describe that substantial antigenic epitopes overlap between a secreted form of HEV ORF2 protein (ORF2S) and other forms of ORF2 proteins and that ORF2S is immunogenic for mice. In addition, they indicated that ORF2S-specific antibody response was detectable in plasma of HEV-infected patients, and that ORF2S can act as a decoy to study B cell response against HEV.

The data indicated in this manuscript are convincing and scientifically sound. However, to improve the present manuscript and strengthen the conclusions, the following concerns should be addressed.

Comments:

1.    Amino acid changes in ORF2 proteins affect the reactivities of anti-ORF2 antibodies against the corresponding epitopes. HEV genotype of E2, p239 and ORF2S (tPA) is described to be 4 or 4a, while that of p495 is 1. To support the authors’ conclusion, alignment of amino acid sequences of these four proteins should be shown in a new figure.

2.    Taxonomical classification of family Hepeviridae has been updated and the virus family is divided into two subfamilies Orthohepevirinae with four genera (Paslahepevirus, Avihepevirus, Rocahepevirus, and Chirohepevirus) and Parahepevirinae with one genus (Piscihepevirus). Therefore, the descriptions on Lines 138-140 should be updated according to https://ictv.global/report/hepeviridae.

3.    Descriptions of HEV genotypes are confusing: for example, genotype 4 (Line 97) and gtA4 (Line 143) should be unified.

4.    If the expressed ORF2S (tPA) protein is glycosylated (Line 222), supporting data should be shown.

5.    No informed consent statement is provided for blood samples from 55 patients infected with HEV (Lines 252-253).

6.    Typographical and grammatical errors are scattered throughout the manuscript and should be carefully revised.

Author Response

Dear reviewer,

Thanks to the editor for arranging the review. Thanks to the reviewers very much for your review of the article "A hepatitis E virus secreted form of ORF2 protein: design strategy, antigenicity and immunogenicity", we have replied the questions raised by reviewers. All changes in the manuscript are highlighted in yellow. Because of your suggestions, the revised manuscript gets better and could get more valuable information for reader. It is hoped that this version of the manuscript can meet the requirements of reviewer.

Yours sincerely
